# Fast rule switching and slow rule updating in a perceptual categorization task

**Flora Bouchacourt**[1†], **Sina Tafazoli**[1†], **Marcelo G Mattar**[1,2], **Timothy J Buschman**[1‡], **Nathaniel D Daw**[1*‡]

[1]Princeton Neuroscience Institute and the Department of Psychology, Princeton, United States; [2]Department of Cognitive Science, University of California, San Diego, San Diego, United States

**Abstract** To adapt to a changing world, we must be able to switch between rules already learned and, at other times, learn rules anew. Often we must do both at the same time, switching between known rules while also constantly re-estimating them. Here, we show these two processes, rule switching and rule learning, rely on distinct but intertwined computations, namely fast inference and slower incremental learning. To this end, we studied how monkeys switched between three rules. Each rule was compositional, requiring the animal to discriminate one of two features of a stimulus and then respond with an associated eye movement along one of two different response axes. By modeling behavior, we found the animals learned the axis of response using fast inference (*rule switching*) while continuously re-estimating the stimulus–response associations within an axis (*rule learning*). Our results shed light on the computational interactions between rule switching and rule learning, and make testable neural predictions for these interactions.

*For correspondence:
ndaw@princeton.edu

[†]These authors contributed equally to this work
[‡]These authors also contributed equally to this work

**Competing interest:** The authors declare that no competing interests exist.

## Editor's evaluation

This important study modeled monkeys' behavior in a stimulus-response rule-learning task to show that animals can adopt mixed strategies involving inference for learning latent states and incremental updating for learning action-outcome associations. The task is cleverly designed, the modeling is rigorous, and importantly there are clear distinctions in the behavior generated by different models, which makes the conclusions convincing.

## Introduction

Intelligence requires learning from the environment, allowing one to modify their behavior in light of experience. A long tradition of research in areas like Pavlovian and instrumental conditioning has focused on elucidating general-purpose trial-and-error learning mechanisms – especially error-driven learning rules associated with dopamine and the basal ganglia (*Daw and O'Doherty, 2014*; *Daw and Shohamy, 2008*; *Daw and Tobler, 2014*; *Dolan and Dayan, 2013*; *Doya, 2007*; *O'Doherty et al., 2004*; *O'Reilly and Frank, 2006*; *Rescorla, 1988*; *Schultz et al., 1997*; *Yin and Knowlton, 2006*; *Day et al., 2007*; *Bayer and Glimcher, 2005*; *Lau and Glimcher, 2008*; *Samejima et al., 2005*; *Padoa-Schioppa and Assad, 2006*). This type of learning works by incremental adjustment that can allow animals to gradually learn an arbitrary task – such as a new stimulus–response discrimination rule. However, in other circumstances, learning can also be quicker and more specialized: for instance, if two different stimulus–response rules are repeatedly reinforced in alternation, animals can come to switch between them more rapidly (*Asaad et al., 1998*; *Rougier et al., 2005*; *Harlow, 1949*).

These more task-specialized dynamics are often modeled by a distinct computational mechanism, such as Bayesian latent state inference, where animals accumulate evidence about which of several 'latent' (i.e., not directly observable) rules currently applies (*Sarafyazd and Jazayeri, 2019*; *Collins and Koechlin, 2012*; *Behrens et al., 2007*; *Gershman et al., 2014*; *Bartolo and Averbeck, 2020*; *Qi et al., 2022*; *Stoianov et al., 2016*). Such inference is associated with activity in the prefrontal cortex (*Durstewitz et al., 2010*; *Milner, 1963*; *Boettiger and D'Esposito, 2005*; *Nakahara et al., 2002*; *Genovesio et al., 2005*; *Boorman et al., 2009*; *Koechlin and Hyafil, 2007*; *Koechlin et al., 2003*; *Sakai and Passingham, 2003*; *Badre et al., 2010*; *Miller and Cohen, 2001*; *Antzoulatos and Miller, 2011*; *Reinert et al., 2021*; *Mansouri et al., 2020*), suggesting that its neural mechanisms are distinct from incremental learning. Lesioning prefrontal cortex impairs performance on tasks that require rule inference (*Milner, 1963*; *Dias et al., 1996*) and neurons in prefrontal cortex track the currently inferred rule (*Mansouri et al., 2006*).

In its simplest form, this type of latent state inference process presupposes that the animals have previously learned about the structure of the task. They must know the set of possible rules, how often they switch, etc. For this reason, latent state inference has typically been studied in well-trained animals (*Sarafyazd and Jazayeri, 2019*; *Asaad et al., 2000*; *White and Wise, 1999*). There has been increasing theoretical interest – but relatively little direct empirical evidence – in the mechanisms by which the brain learns the broader structure of the task in order to build task-specialized inference mechanisms for rapid rule switching. For Bayesian latent state inference models, this problem corresponds to learning the generative model of the task, for example inferring a mixture model over latent states (rules or task conditions) and their properties (e.g., stimulus–response–reward contingencies) (*Sarafyazd and Jazayeri, 2019*; *Collins and Koechlin, 2012*; *Collins and Frank, 2016*; *Schuck et al., 2016*; *Chan et al., 2016*; *Hampton et al., 2006*; *Frank and Badre, 2012*; *Purcell and Kiani, 2016*). In principle, this too could be accomplished by hierarchical Bayesian inference over which of several already specified rules currently applies and over the space of possible rules, as in Chinese restaurant process models. However, such hierarchical inference is less tractable and hard to connect to neural mechanisms.

Perhaps most intriguingly, one solution to the lack of a full process-level model of latent state learning is that rule learning and rule switching involve an interaction between both major classes of learning mechanisms – latent state inference and incremental trial-and-error learning. Thus, in Bayesian inference models, it is often hypothesized that an inferential process decides which latent state is in effect (e.g., in prefrontal cortex), while the properties of each state are learned, conditional on this, by downstream error-driven incremental learning (e.g, in the striatum) (*Padoa-Schioppa and Assad, 2006*; *Collins and Koechlin, 2012*; *Frank and Badre, 2012*; *Seo et al., 2012*; *Rushworth et al., 2011*; *Balewski et al., 2022*). However, these two learning mechanisms have mostly been studied in regimes where they operate in isolation (*Sarafyazd and Jazayeri, 2019*; *Asaad et al., 2000*; *White and Wise, 1999*; *Lak et al., 2020*; *Busse et al., 2011*; *Fründ et al., 2014*; *Gold et al., 2008*; *Tsunada et al., 2019*) and, apart from a few examples in human rule learning (*Collins and Koechlin, 2012*; *Collins and Frank, 2016*; *Donoso et al., 2014*; *Badre and Frank, 2012*; *Bouchacourt et al., 2020*; *Franklin and Frank, 2018*), their interaction has been limited to theoretical work.

To study rule switching and rule learning, we trained non-human primates to perform a rule-based category-response task. Depending on the rule in effect, the animals needed to attend to and categorize either the color or the shape of a stimulus, and then respond with a saccade along one of two different response axes. We observe a combination of both fast and slow learning during the task: monkeys rapidly switched into the correct response axis, consistent with inferential learning of the response state, while, within a state, the animals slowly learned category-response mappings, consistent with incremental (re)learning. This was true even though the animals were well trained on the task beforehand. To quantify the learning mechanisms underlying the animals' behavior, we tested whether inference or incremental classes of models, separately, could explain the behavior. Both classes of models reproduced learning-like effects – that is dynamic, experience-driven changes in behavior. However, neither model could, by itself, explain the combination of both fast and slow learning. Importantly, the fact that a fully informed inferential learner could not explain the behavior also indicated that the observed fast and slow learning was not simply driven by informed adjustment to the task structure itself. Instead, we found that key features of behavior were well explained by a hybrid rule-switching and rule-learning model, which inferred which response axis was active while

continually performing slower, incremental relearning of the consequent stimulus–response mappings within an axis. These results support the hypothesis that there are multiple, interacting, mechanisms that guide behavior in a contextually appropriate manner.

## Results

### Task design and performance

Two rhesus macaques were trained to perform a rule-based category-response task. On each trial, the monkeys were presented with a stimulus that was composed of a color and shape (*Figure 1a*). The shape and color of each stimulus were drawn from a continuous space and, depending on the current rule in effect, the animals categorized the stimulus according to either its color (red vs. green) or its shape ('bunny' vs.'tee', *Figure 1b*). Then, as a function of the category of the stimulus and the current rule, the animals made one of four different responses (an upper-left, upper-right, lower-left, or lower-right saccade).

Animals were trained on three different category-response rules (*Figure 1c*). Rule 1 required the animal to categorize the shape of the stimulus, making a saccade to the upper-left location when the shape was categorized as a 'bunny' and a saccade to the lower-right location when the shape was categorized as a 'tee'. These two locations – upper-left and lower-right – formed an 'axis' of response (*Axis 1*). Rule 2 was similar but required the animal to categorize the color of the stimulus and then respond on the opposite axis (*Axis 2*; red = upper-right, green = lower-left). Finally, Rule 3 required categorizing the color of the stimulus and responding on *Axis 1* (red = lower-right, green = upper-left). Note that these rules are compositional in nature, with overlapping dimensions (*Figure 1d*). Rule 1 required categorizing the shape of the stimulus, while Rules 2 and 3 required categorizing the color of the stimulus. Similarly, Rules 1 and 3 required responding on the same axis (*Axis 1*), while Rule 2 required a different set of responses (*Axis 2*). In addition, the overlap in the response axis for Rules 1 and 3 meant certain stimuli had congruent responses for both rules (e.g., red-tee and green-bunny stimuli) while other stimuli had incongruent responses between rules (e.g., red-bunny and green-tee). For all rules, when the animal made a correct response, it received a reward (an incorrect response led to a short 'time-out').

Animals performed the same rule during a block of trials. Critically, the animals were not explicitly cued as to which rule was in effect for that block. Instead, they had to use information about the stimulus, their response, and reward feedback, to infer which rule was in effect. After the animals discovered the rule and were performing it at a high level (defined as >70%, see Methods) the rule would switch. Although unpredictable, the moment of switching rules was cued to the animals (with a flashing screen). Importantly, this switch cue did not indicate which rule was now in effect (just that a switch had occurred).

To facilitate learning and performance, the sequence of rules across blocks was semi-structured such that the axis of response always changed following a block switch. This means that the animals always alternated between a Rule 2 block and either a Rule 1 or 3 block (chosen pseudo randomly), and that Rule 2 thus occurred twice as frequently as the other rules. Note that the cue implied an axis switch but did not instruct which axis to use, thus the animals must learn and continually track on which axis to respond.

Overall, both monkeys performed the task well above chance (*Figure 1e, f*). When the rule switched to Rule 2, the animals quickly switched their behavior: Monkey S responded correctly on the first trial in 81%, confidence interval (CI) = [0.74,0.87] of Rule 2 blocks, and reached 91%, CI = [0.85,0.95] after only 20 trials (Monkey C being, respectively, at 78%, CI = [0.65,0.88]; and 85%, CI = [0.72,0.92]). In Rules 1 and 3, their performance also exceeded chance level quickly. In Rule 1, although the performance of Monkey S was below chance on the first trial (0%, CI = [0,0.052]; 46%, CI = [0.28,0.65] for Monkey C), reflecting perseveration on the previous rule, performance quickly climbed above chance (77% after 50 trials, CI = [0.66,0.85]; 63%, CI = [0.43,0.79] for Monkey C). A similar pattern was seen for Rule 3 (initial performance of 1.5%, CI = [0.0026,0.079] and 78%, CI = [0.67,0.86] after 50 trials for Monkey S; 41%, CI = [0.25,0.59] and 67%, CI = [0.48,0.81] for Monkey C, respectively).

While the monkeys performed all three rules well, there were two interesting behavioral phenomena. First, the monkeys were slower to switch to Rules 1 and 3 than to switch to Rule 2. On the first 20 trials, the difference in average percent performance of Monkey S was $\Delta = 35$ between Rules 2 and 1, and $\Delta$

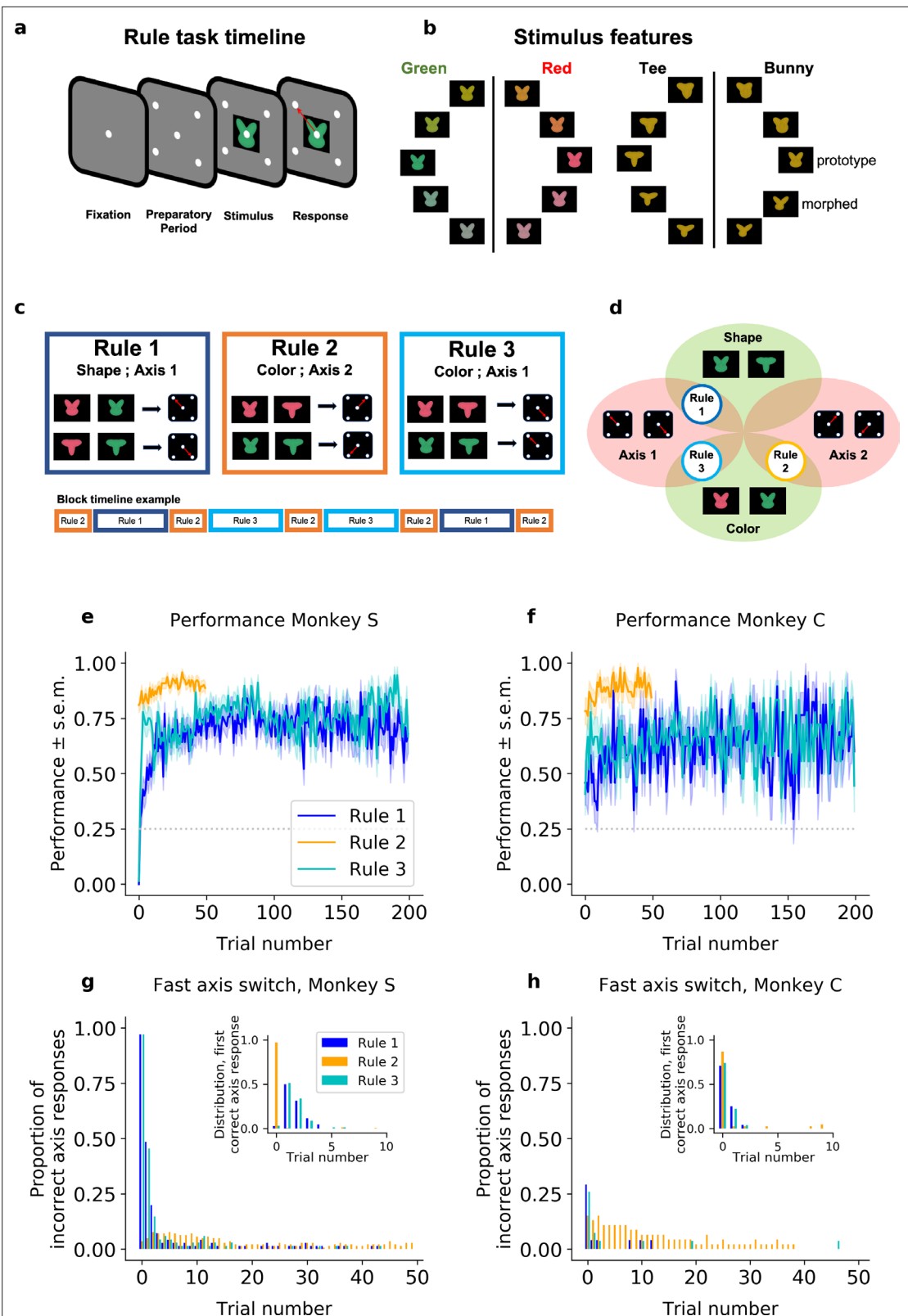

**Figure 1.** Task design and performance (including all trials). (**a**) Schematic of a trial. (**b**) Stimuli were drawn from a two-dimensional feature space, morphing both color (left) and shape (right). Stimulus categories are indicated by vertical lines and labels. (**c**) The stimulus–response mapping for the three rules, and an example of a block timeline. (**d**) Venn diagram showing the overlap between rules. Average performance (sample mean and standard error of the mean) for each rule, for (**e**) Monkey S and (**f**) Monkey C. Proportion of responses on the incorrect axis for the first 50 trials of each block for (**g**) Monkey S and (**h**) Monkey C. Insets: Trial number of the first response on the correct axis after a block switch, respectively, for Monkeys S and C.

= 22 between Rules 2 and 3 (significant Fisher's test comparing Rule 2 to Rules 1 and 3, with p < 10⁻⁴ in both conditions; respectively, $\Delta$ = 35, $\Delta$ = 24 and p < 10⁻⁴ for Monkey C).

Second, both monkeys learned the axis of response nearly instantaneously. After a switch cue, Monkey S almost always responded on Axis 2 (the response axis consistent with Rule 2; 97%, CI = [0.90,0.99] in Rule 1; 97%, CI = [0.92,0.98] in Rule 2; 97%, CI = [0.90,0.99] in Rule 3; see *Figure 1g*). Then, if this was incorrect, it switched to the correct axis within five trials on 97%, CI = [0.90,0.99] of blocks of Rule 1, and 94% CI = [0.86,0.98] of blocks of Rule 3. Monkey C instead tended to alternate the response axis on the first trial following a switch cue (it made a response on the correct axis on the first trial with a probability of 71%, CI = [0.51,0.85] in Rule 1; 85%, CI = [0.72,0.92] in Rule 2; and 84%, CI = [0.55,0.87] in Rule 3), implying an understanding of the pattern of axis changes with block switches (*Figure 1h*). Both monkeys maintained the correct axis with very few off-axis responses throughout the block (at trial 20, Monkey S: 1.4%, CI = [0.0025,0.077] in Rule 1; 2.1%, CI = [0.0072,0.060] in Rule 2; 0%, CI = [0,0.053] in Rule 3; Monkey C: 0%, CI = [0,0.14] in Rule 1; 4.3%, CI = [0.012,0.15] in Rule 2; 3.7%, CI = [0.0066,0.18] in Rule 3). These results suggest the animals were able to quickly identify the axis of response but took longer (particularly for Rules 1 and 3) to learn the correct mapping between stimulus features and responses within an axis.

## Learning rules de novo cannot capture the behavior

To perform the task, the animals had to learn which rule was in effect during each block of trials. This required determining both the response axis and the relevant feature. As noted above, the monkeys' behavior suggests learning was a mixture of fast switching, reminiscent of inference models, and slow refinement, as in error-driven incremental learning. Given this, we began by testing whether the inference or incremental classes of models could capture the animals' behavior. As in previous work (*Dayan and Daw, 2008*; *Pouget et al., 2016*; *Pouget et al., 2013*; *Gold and Shadlen, 2001*; *Bichot and Schall, 1999*), all our models shared common noisy perceptual input and action selection stages (*Figure 2—figure supplement 1*, and Methods). As we detail next, the intervening mechanism for mapping stimulus to action value differed between models.

First, we fit an error-driven learning model, which gradually relearns the stimulus–response mappings de novo at the start of each block, to the monkeys' behavior. This class of models works by learning the reward expected for different stimulus–response combinations, using incremental running averages to smooth out trial-to-trial stochasticity in reward realization – here, due to perceptual noise in the stimulus classification. In particular, we fit a variant of Q learning (model QL, see

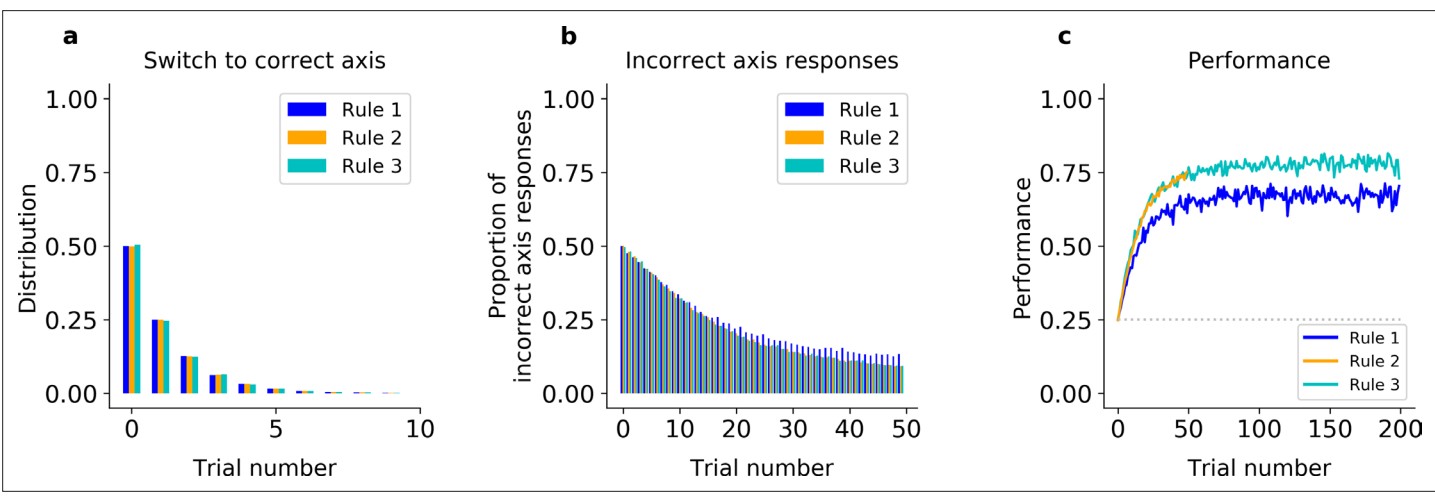

**Figure 2.** Incremental learner (QL) model fitted on Monkey S behavior (see *Figure 2—figure supplement 2* for Monkey C). (**a**) Trial number of the first response of the model on the correct axis after a block switch (compare to *Figure 1g*, inset). (**b**) Proportion of responses of the model on the incorrect axis for the first 50 trials of each block (compare to *Figure 1g*). (**c**) Model performance for each rule (averaged over blocks, compare to *Figure 1e*).

The online version of this article includes the following figure supplement(s) for figure 2:

**Figure supplement 1.** The three models.

**Figure supplement 2.** Incremental learner (QL) model fitted on Monkey C behavior (see *Figure 2* for Monkey S).

*Figure 2—figure supplement 1* and Methods) that was elaborated to improve its performance in this task: for each action, model QL parameterized the mapping from stimulus to reward linearly using two basis functions over the feature space (one binary indicator each for color and shape), and used error-driven learning to estimate the appropriate weights on these for each block. This scheme effectively builds-in the two relevant feature-classification rules (shape and color), making the generous assumption that the animals had already learned the categories. In addition, the model resets the weights to fixed initial values at each block switch, allowing the model to start afresh and avoiding the need to unlearn. Yet, even with these built-in advantages, the model was unable to match the animals' ability to rapidly switch axes, but instead relearned the feature–response associations after each block switch (simulations under best-fitting parameters shown in *Figure 2* for Monkey S, *Figure 2—figure supplement 2* for Monkey C). Several tests verified the model learned more slowly than the animals (*Figure 2a, b*). For instance, the model fitted on Monkey S's behavior responded on Axis 2 on the first trial of the block only 50% of the time in all three rules (*Figure 2a*, Fisher's test of model simulations against data: $p < 10^{-4}$ for the three rules). The model thus failed to capture the initial bias of Monkey S for Rule 2 discussed above. Importantly, the model switched to the correct axis within five trials on only 58% of blocks of Rule 1, and 57% of blocks of Rule 3 (*Figure 2b*, Fisher's test against monkey behavior: $p < 10^{-4}$). Finally, the model performed 24% of off-axis responses after 20 trials in Rule 1, 21% in Rule 2, and 21% in Rule 3, all much higher than what was observed in the monkey's behavior (Fisher's test $p < 10^{-4}$).

In addition, because of the need to relearn feature–response associations after each block switch, the incremental QL model was unable to capture the dichotomy between the monkeys' slower learning in Rules 1 and 3 (which share Axis 1) and the faster learning of Rule 2 (using Axis 2). As noted above, the monkeys performed Rule 2 at near asymptotic performance from the beginning of the block but were slower to learn which feature to attend to on blocks of Rules 1 and 3 (*Figure 1g, h*). In contrast, because the model learned each rule in the same way, the incremental learner performed similarly on all three rules (*Figure 2c* for Monkey S, *Figure 2—figure supplement 2c* for Monkey C). In particular, it performed correctly on the first trial in only 25% of Rule 2 blocks (Fisher's test against behavior: $p < 10^{-4}$), and reached only 62% after 20 trials ($p < 10^{-4}$). As a result, on the first 20 trials, the difference in average percent performances was only $\Delta = 4.0$ between Rules 2 and 1, and was only $\Delta = 0.76$ between Rules 2 and 3 (similar results were seen when fitting the model to Monkey C, see *Figure 2—figure supplement 2*). The same pattern of results was seen when the initial weights were free parameters (see Methods).

Altogether, these results argue simple incremental relearning of the axes and features de novo cannot reproduce the animals' ability to instantaneously relearn the correct axis after a block switch or the observed differences in learning speed between the rules.

## Pure inference of previously learned rules cannot capture the behavior

The results above suggest that incremental learning is too slow to explain the quick switch between response axes displayed by the monkeys. So, we tested whether a model that leverages Bayesian inference can capture the animals' behavior. A fully informed Bayesian ideal observer model (IO, see *Figure 2—figure supplement 1* and Methods) uses statistical inference to continually estimate which of the three rules is in effect, accumulating evidence ('beliefs') for each rule based on the history of previous stimuli, actions, and rewards. The IO model chooses the optimal action for any given stimulus, by averaging the associated actions' values under each rule, weighted by the estimated likelihood that each rule is in effect. Like incremental learning, the IO model learns and changes behavior depending on experience. However, unlike incremental models, this model leverages perfect knowledge of the rules to learn rapidly, limited only by stochasticity in the evidence. Here, noisy stimulus perception is the source of such stochasticity, limiting both the speed of learning and asymptotic performance. Indeed, the IO model predicts the speed of initial (re)learning after a block switch should be coupled to the asymptotic level of performance. Furthermore, given that perceptual noise is shared across rules, the IO model also predicts the speed of learning will be the same for rules that use the same features.

As expected, when fit to the animal's behavior, the IO model reproduced the animals' ability to rapidly infer the correct axis (*Figure 3—figure supplement 1a, b, d, e*). For example, when fitted to Monkey S behavior, the model initially responded on Axis 2 almost always immediately after each

block switch cue (96% in all rules, Fisher's test against monkey's behavior p > 0.4). Then, if this was incorrect, the model typically switched to the correct axis within five trials on 89% of blocks of Rule 1, and 95% of blocks of Rule 3 (Fisher's test against monkey behavior: p > 0.2 in both rules). The model maintained the correct axis with very few off-axis responses throughout the block (after trial 20, 1.3% in Rule 1; 1.1% in Rule 2; 1.2% in Rule 3; Fisher's test against monkey's behavior: p > 0.6 in all rules).

However, the IO model could not capture the observed differences in learning speed for the different rules (*Figure 3—figure supplement 1c, f*). To understand why, we looked at performance as a function of stimulus difficulty. As expected, the monkey's performance depended on how difficult it was to categorize the stimulus (i.e., the morph level; psychometric curves shown in *Figure 3a–c* for Monkey S, *Figure 3—figure supplement 2a–c* for Monkey C). For example, in color blocks (Rules 2 and 3), the monkeys performed better for a 'prototype' red stimulus than for a 'morphed' orange stimulus (*Figure 3a–c*). Indeed, on 'early trials' (first 50 trials) of Rule 2, Monkey S correctly responded to 96% (CI = [0.95,0.97]) of prototype stimuli, and only to 91%, CI = [0.90,0.92] of 'morphed' stimuli (p < 10⁻⁴; similar results for Monkey C in *Figure 3—figure supplement 2*). Rule 3 had a similar ordering: Monkey S correctly responded to 80% (CI = [0.77,0.82]) of prototype stimuli, and only 62%, CI = [0.60,0.64] of 'morphed' stimuli (p < 10⁻⁴). This trend continued as the animal learned Rule 3 (trials 50–200; 89%, CI = [0.88,0.90] and 74%, CI = [0.73,0.75], respectively, for prototype and morphed stimuli, p < 10⁻⁴).

Importantly, there was a discrepancy between the performance on 'morphed' stimuli in Rule 2 versus Rule 3, with a difference in average percent performance of $\Delta$ = 28 for the first 50 trials in both rules (p < 10⁻⁴). This was still true, even if we considered Rule 2 against the last trials of Rule 3 ($\Delta$ = 17, p < 10⁻⁴). The same discrepancy was observed between the performance on 'prototype' stimuli in Rule 2 versus Rule 3, with a difference in average percent performance on $\Delta$ = 16 for the first 50 trials in both rules (p < 10⁻⁴), and $\Delta$ = 6.8 if comparing to the last trials of Rule 3 (p < 10⁻⁴).

The IO model captured the performance ordering on morphed and prototype stimuli for each rule (*Figure 3d–i*, similar results for the model reproducing Monkey C, *Figure 3—figure supplement 2*). However, the model performed similarly for morphed stimuli on Rules 2 and 3. This is because both rules involve categorizing color and so they shared the same perceptual noise, leading to the same likelihood of errors. Furthermore, because perceptual noise limits learning and asymptotic performance in the IO model, it predicts the speed of learning should be shared across Rules 2 and 3, and initial learning in both rules on the first 50 trials should be coupled to the asymptotic performance. Given this, the model had to trade-off between behavioral performance in Rules 2 and 3. Using best-fit parameters, the model reproduced the animals' lower asymptotic performance in Rule 3 by increasing color noise, and so it failed to capture the high performance on Rule 2 early on (*Figure 3e, f*, *Figure 3—figure supplement 2e, f*). The resulting difference in average percent performance for 'morphed' stimuli was only $\Delta$ = 4.0 for the first 50 trials and $\Delta$ = 0.0044 if we considered the last trials of Rule 3 (respectively, $\Delta$ = 4.2 and $\Delta$ = 0.032 for 'prototype'). Conversely, if we forced the model to improve color perception (by reducing perceptual noise, *Figure 3h, i*, and *Figure 3—figure supplement 2h, i*), then it was able to account for the monkeys' performance on Rule 2, but failed to match the animals' behavior on Rule 3. The resulting difference in average percent performance was again only $\Delta$ = 3.4 for the first 50 trials, and $\Delta$ = −0.17 if we considered the last trials of Rule 3 (respectively, $\Delta$ = 3.2 and $\Delta$ = −0.16 for 'prototype').

One might be concerned that including a correct generative prior on the transition between axes given by the specific task structure would solve this issue, as a Rule 2 block is always following a Rule 1 or 3 block, hence possibly creating an inherent discrepancy in learning Rule 2 versus Rules 1 and 3. However, the limiting factor was not the speed for axis discovery (which was nearly instantaneous, cf above), but the shared perceptual color noise between Rules 2 and 3, coupling initial learning to asymptotic performance. Such a modified IO could not account for the behavior (*Figure 3—figure supplement 3*).

## The key features of monkeys' behavior are reproduced by a hybrid model composing inference over axes and incremental relearning over features

To summarize, the main characteristics of the animals' behavior were (1) rapid learning of the axis of response after a block switch, (2) immediately high behavioral performance of Rule 2, the only rule

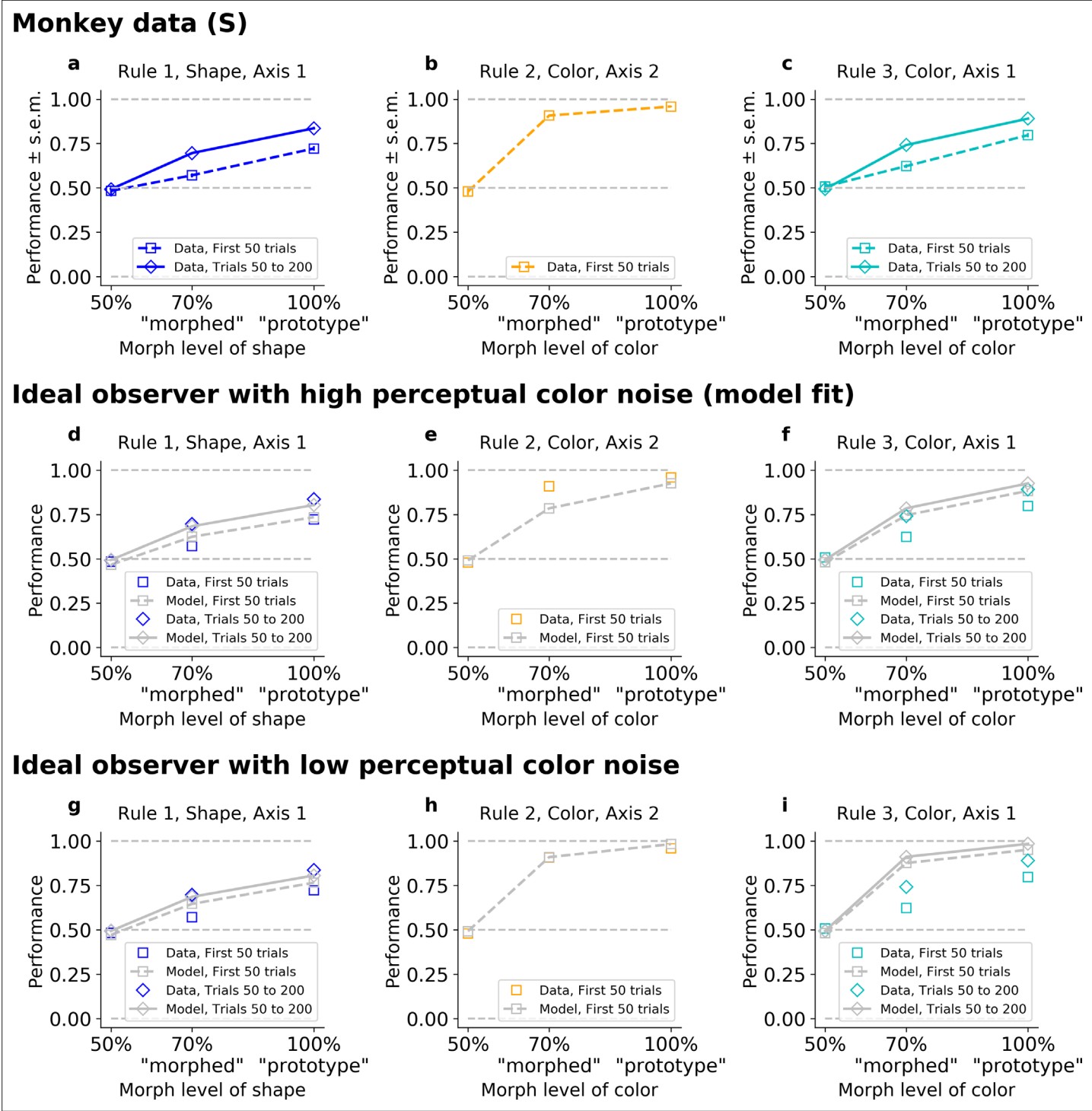

**Figure 3.** The ideal observer (IO), slow or fast, but not both. Fitted on Monkey S behavior (see *Figure 3—figure supplement 1* for Monkey C). Note that data were collapsed across 50%/150%; 30%/70%/130%/170%; and 0%/100% (non-collapsed psychometric functions can be seen in *Figure 5*). (**a–c**) Performance for Rules 1, 2, and 3, as a function of the morphed version of the relevant feature. (**d–f**) Performance for Rules 1, 2, and 3, for IO model with high color noise. This parameter regime corresponds to the case where the model is fitted to the monkey's behavior (see Methods). (**g–i**) Performance for Rules 1, 2, and 3, for IO model with low color noise. Here, we fixed $\kappa_C = 6$.

The online version of this article includes the following figure supplement(s) for figure 3:

**Figure supplement 1.** Ideal observer (IO) model fitted on Monkeys S and C.

**Figure supplement 2.** The ideal observer (IO), slow or fast, but not both.

**Figure supplement 3.** The ideal observer model including a correct generative prior on the transition between axes given by the specific task structure.

on Axis 2, and (3) slower relearning of Rules 1 and 3, which mapped different features onto Axis 1. Altogether, these results suggest that the animals learned axes and features separately, with fast learning of the axes and slower learning of the features. One way to conceive this is as a Bayesian inference model (similar to IO), but relaxing the assumption that the animal had perfect knowledge of the underlying rules (i.e., all of the stimulus–action–reward contingencies). We propose that the animals maintained two latent states (e.g., one corresponding to each axis of response) instead of the three rules we designed. Assuming each state had its own stimulus–action–reward mappings, the mappings would be stable for Rule 2 (Axis 2) but continually re-estimated for Rules 1 and 3 (Axis 1). To test this hypothesis, we implemented a hybrid model that inferred the axis of response while incrementally learning which features to attend for that response axis (Hybrid Q Learner, 'HQL' in the Methods, *Figure 2—figure supplement 1*). In the model, the current axis of response was inferred through Bayesian evidence accumulation (as in the IO model), while feature–response weights were incrementally learned for each axis of response.

Intuitively, this model could explain all three core behavioral observations. First, inference allows for rapid switching between axes. Second, because only Rule 2 mapped to Axis 2, the weights for Axis 2 did not change and so the model was able to perform well on Rule 2 immediately. Third, because Rules 1 and 3 shared an axis of response, and, thus, a single set of feature–response association

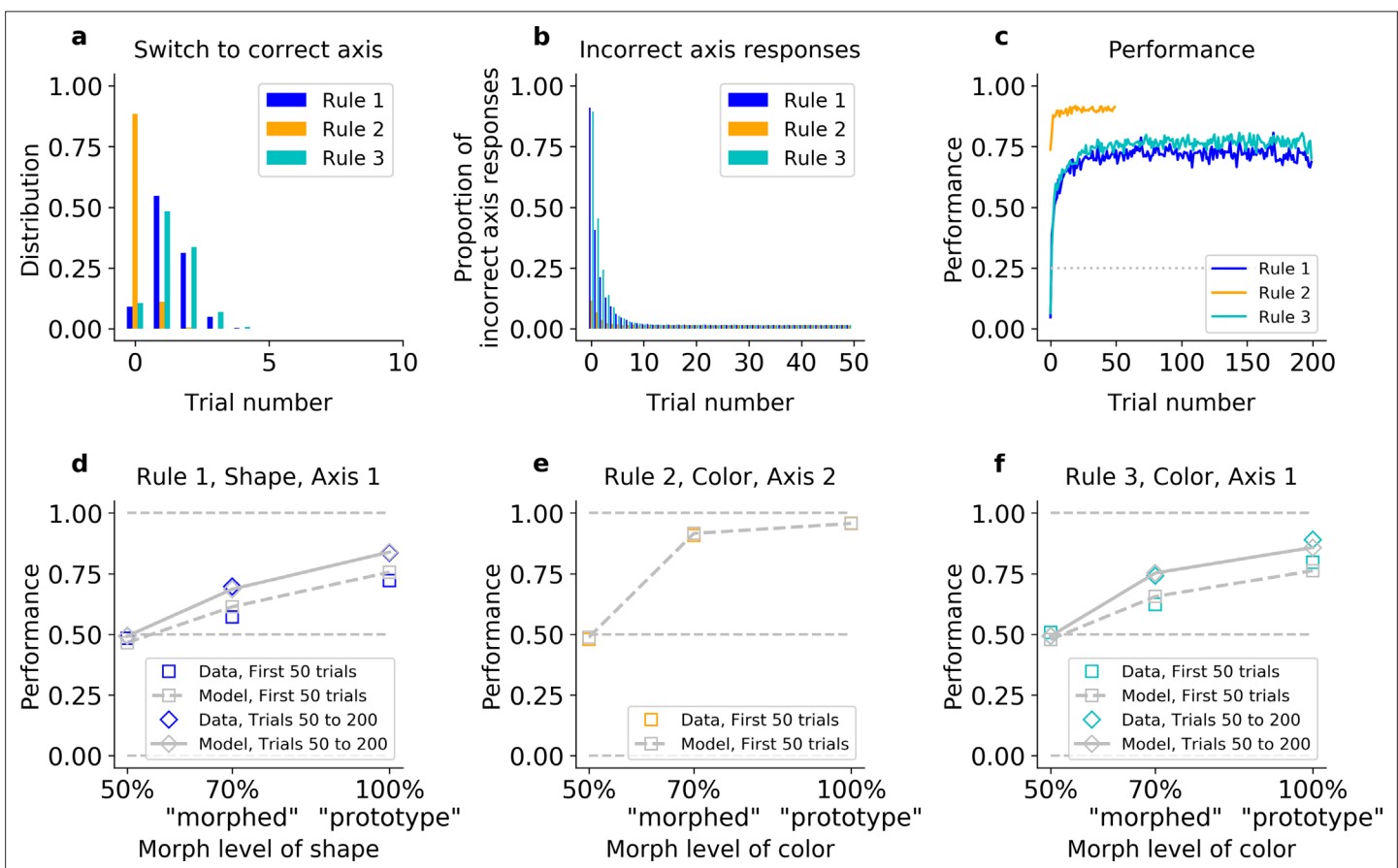

**Figure 4.** The hybrid learner (HQL) accounts both for fast switching to the correct axis, and slow relearning of Rules 1 and 3. Model fit on Monkey S, see *Figure 4—figure supplement 1* for Monkey C. (**a**) Trial number for the first response on the correct axis after a block switch, for the model (compare to *Figure 1e* inset). (**b**) Proportion of responses on the incorrect axis for the first 50 trials of each block, for the model (compare to *Figure 1e*). (**c**) Performance of the model for the three rules (compare to *Figure 1g*). (**d–f**) Performance for Rules 1, 2, and 3, as a function of the morphed version of the relevant feature.

The online version of this article includes the following figure supplement(s) for figure 4:

**Figure supplement 1.** The hybrid learner (HQL) accounts both for fast switching to the correct axis, and slow relearning of Rules 1 and 3.

**Figure supplement 2.** The hybrid learner, beliefs, and weights.

**Table 1.** Models parameters.

| Monkey S | Noise perception | | Learning rate | Initial belief R1 | Initial belief R3 | Initial belief Axis 1 | Weight decay | Initial weights |
|---|---|---|---|---|---|---|---|---|
| | $\kappa$ (color) | $\kappa$ (shape) | $\alpha$ | $b_1$ | $b_3$ | bax | $\eta$ | $w_0$ |
| QL model | Mean = 2.2 std = 0.44 | Mean = 1.3 std = 0.27 | Mean = 0.23 std = 0.039 | | | | | |
| IO model | Mean = 2.4 std = 0.46 | Mean = 1.3 std = 0.31 | | Mean = 0.091 std = 0.089 | Mean = 0.14 std = 0.10 | | | |
| HQL model | Mean = 11 std = 1.2 | Mean = 5.1 std = 2.2 | Mean = 0.23 std = 0.10 | | | Mean = 0.29 std = 0.076 | Mean = 0.046 std = 0.022 | Mean = [−0.61,0.79,0.77,−0.64, −0.83,0.035,0.74,0.040] std = [0.23,0.089,0.13,0.17, 0.053,0.59,0.19,0.57] |

| Monkey C | Noise perception | | Learning rate | Initial belief R1 | Initial belief R3 | Initial belief Axis 1 | Weight decay | Initial weights |
|---|---|---|---|---|---|---|---|---|
| | $\kappa$ (color) | $\kappa$ (shape) | $\alpha$ | $b_1$ | $b_3$ | bax | $\eta$ | $w_0$ |
| QL model | Mean = 1.2 std = 0.13 | Mean = 0.71 std = 0.090 | Mean = 0.18 std = 0.010 | | | | | |
| IO model | Mean = 1.3 std = 0.21 | Mean = 0.71 std = 0.18 | | Mean = 0.35 std = 0.16 | Mean = 0.32 std = 0.16 | | | |
| HQL model | Mean = 12 std = 2.4 | Mean = 7.0 std = 3.4 | Mean = 0.12 std = 0.10 | | | Fixed to 0.5 | Mean = 0.067 std = 0.060 | Mean = [−0.45,0.56,0.60,−0.56, −0.83,−0.12,0.70,−0.19] std = [0.16,0.12,0.093,0.16, 0.051,0.45,0.25,0.41] |

weights, this necessitated relearning associations for each block, reflected in the animal's slower learning for these rules.

Consistent with this intuition, the HQL model provided an accurate account of the animals' behavior. First, unlike the QL model, the HQL model reproduced the fast switch to the correct axis (*Figure 4a, b*, *Figure 4—figure supplement 1a, bCited*, and *Figure 4—figure supplement 2a–c and g–i*). Fitted to Monkey S behavior, the model initially responded on Axis 2 immediately after each block switch cue (91% in Rule 1, 89% in Rules 2 and 3, Fisher's test against monkey's behavior p > 0.05). Then, if this was incorrect, the model switched to the correct axis within five trials on 91% of blocks of Rules 1 and 3 (Fisher's test against behavior: p > 0.2 in both rules). Similar to the animals, the model maintained the correct axis with very few off-axis responses throughout the block (on trial 20, 1.4% in Rule 1; 1.3%, in Rule 2; 1.5% in Rule 3, Fisher's test against monkey's behavior: p > 0.7 in all rules).

Second, contrary to the IO model, the HQL model captured the animals' fast performance on Rule 2 and slower performance on Rules 1 and 3 (*Figure 4c* and *Figure 4—figure supplement 1c*). As detailed above, animals were significantly better on Rule 2 than Rules 1 and 3 on the first 20 trials. The model captured this difference: fitted on Monkey S's behavior, the difference in average percent performance on the first 20 trials was $\Delta$ = 31 between Rules 2 and 1, and $\Delta$ = 29 between Rules 2 and 3 (a Fisher's test against monkey's behavior gave p > 0.05 for the first trial, p > 0.1 for trial 20).

Third, the HQL model captured the trade-off between the animals' initial learning rate and asymptotic behavioral performance in Rules 2 and 3 (*Figure 4d–f* and *Figure 4—figure supplement 1d–f*). Similar to the animals, the resulting difference in average percent performance for 'morphed' stimuli was $\Delta$ = 26 for the first 50 trials ($\Delta$ = 16 if we considered the last trials of Rule 3; $\Delta$ = 19 and $\Delta$ = 9.9 for early and late 'prototype' stimuli, respectively). The model was able to match the animals' performance because the weights for Axis 2 did not change from one Rule 2 block to another (*Figure 4— figure supplement 2e, k*), and the estimated perceptual noise of color was low in order to account for the high performance of both morphed and prototype stimuli (*Figure 4e* and *Figure 4—figure supplement 1e*). To account for the slow re-learning observed for Rules 1 and 3, the best-fitting learning rate for feature–response associations was relatively low (*Figure 4d, f* and *Figure 4—figure supplement 1d, f*, *Figure 4—figure supplement 2d, f, j, l*, *Table 1*).

## The effect of stimulus congruency (and incongruency) provides further evidence for the hybrid model

To further understand how the HQL model outperforms the QL and IO models, we examined the animal's behavioral performance as a function of the relevant and irrelevant stimulus features. The orthogonal nature of the features and rules meant that stimuli could fall into two general groups: congruent stimuli had features that required the same response for both Rules 1 and 3 (e.g., a green bunny, *Figure 1*) while incongruent stimuli had features that required opposite responses between the two rules (e.g., a red bunny). Consistent with previous work (*Noppeney et al., 2010*; *Venkatraman et al., 2009*; *Bugg et al., 2008*; *Carter et al., 1995*; *Musslick and Cohen, 2021*), the animals performed better on congruent stimuli than incongruent stimuli (*Figure 5a* for Monkey S, *Figure 5—figure supplement 1a* for Monkey C). This effect was strongest during learning, but persisted throughout the block (*Figure 5—figure supplement 2a, e*): during early trials of Rules 1 and 3, the monkeys' performance was significantly higher for congruent stimuli than for incongruent stimuli (gray vs. red squares in *Figure 5b*; 94%, CI = [0.93,0.95] vs. 57%, CI = [0.55,0.58], respectively; $\Delta$ = 37, Fisher's test p < $10^{-4}$; see *Figure 5—figure supplement 1b* for Monkey C). Similarly, the animals were slower to respond to incongruent stimuli (*Figure 5—figure supplement 3*, $\Delta$ = 25 ms in reaction time for incongruent and congruent stimuli, *t*-test, p < $10^{-4}$). In contrast, the congruency of stimuli had no effect during Rule 2 – behavior depended only on the stimulus color, suggesting the monkeys ignored the shape of the stimulus during Rule 2, even when the morph level of the color was more difficult (gray vs. red squares in *Figure 5c*; performance was 92%, CI = [0.90,0.93], and 93%, CI = [0.92,0.93] for congruent and incongruent stimuli, respectively; with $\Delta$ = −0.73; Fisher's test, p = 0.40; see *Figure 5—figure supplement 1c* for Monkey C).

This incongruency effect provided further evidence for the HQL model. First, pure incremental learning by the QL model did not capture this result, but instead predicted an opposite effect. This is because incongruent trials were four times more likely than congruent trials (see Methods). As the QL model encodes the statistics of the task through error-driven updating of action values, the proportion of congruent versus incongruent trials led to an anti-incongruency effect – the QL model fit to Monkey S predicted worse performance on congruent than incongruent trials (*Figure 5d, e*; 45% and 62%, respectively; $\Delta$ = −16; Fisher's test p < $10^{-4}$; see *Figure 5—figure supplement 1d, e* for Monkey C). Furthermore, for the same reason, the QL model produced a difference in performance during Rule 2 (*Figure 5f*; 54% for congruent vs. 64% for incongruent; $\Delta$ = −10; Fisher's test p < $10^{-4}$; see *Figure 5—figure supplement 1f* for Monkey C, see also *Figure 5—figure supplement 2b, f* for this effect throughout the block).

Second, the IO model also did not capture the incongruency effect. In principle, incongruency effects can be seen in this type of model when perceptual noise is large, because incongruent stimuli are more ambiguous when the correct rule is not yet known. However, given the statistics of the task, learning in the IO model quickly reached asymptotic performance, for both congruent and incongruent trials (*Figure 5g, h*; 75% and 72%, respectively; $\Delta$ = 3.8 only; *Figure 5—figure supplement 1g, h* for Monkey C), hence not reproducing the incongruency effect.

In contrast to the QL and IO models, the hybrid HQL model captured the incongruency effect. In the HQL model, the weights for mapping congruent stimuli to responses were the same for Rules 1 and 3. In contrast, the weights for incongruent stimuli must change for Rules 1 and 3. Therefore, the animals' performance was immediately high on congruent stimuli, while the associations for incongruent stimuli had to be relearned on each block (*Figure 5—figure supplement 4*). The model fitted to Monkey S behavior reproduced the greater performance on congruent than incongruent stimuli (*Figure 5g, h*; 92% and 61%, respectively, $\Delta$ = 31; see *Figure 5—figure supplement 1g, h* for Monkey C). As with the monkey's behavior, this effect persisted throughout the block (*Figure 5—figure supplement 2d, h*). Finally, the HQL model captured the absence of incongruency effect in Rule 2 (*Figure 5i*, green vs. red squares; 92% and 93%, respectively; $\Delta$ = −1.0; see *Figure 5—figure supplement 1i* for Monkey C), as there was no competing rule, there was no need to update the Axes 2 weights between blocks. As a result, only a hybrid model performing both rule switching of axis and rule learning of features could account for the incongruency effect observed in the behavior.

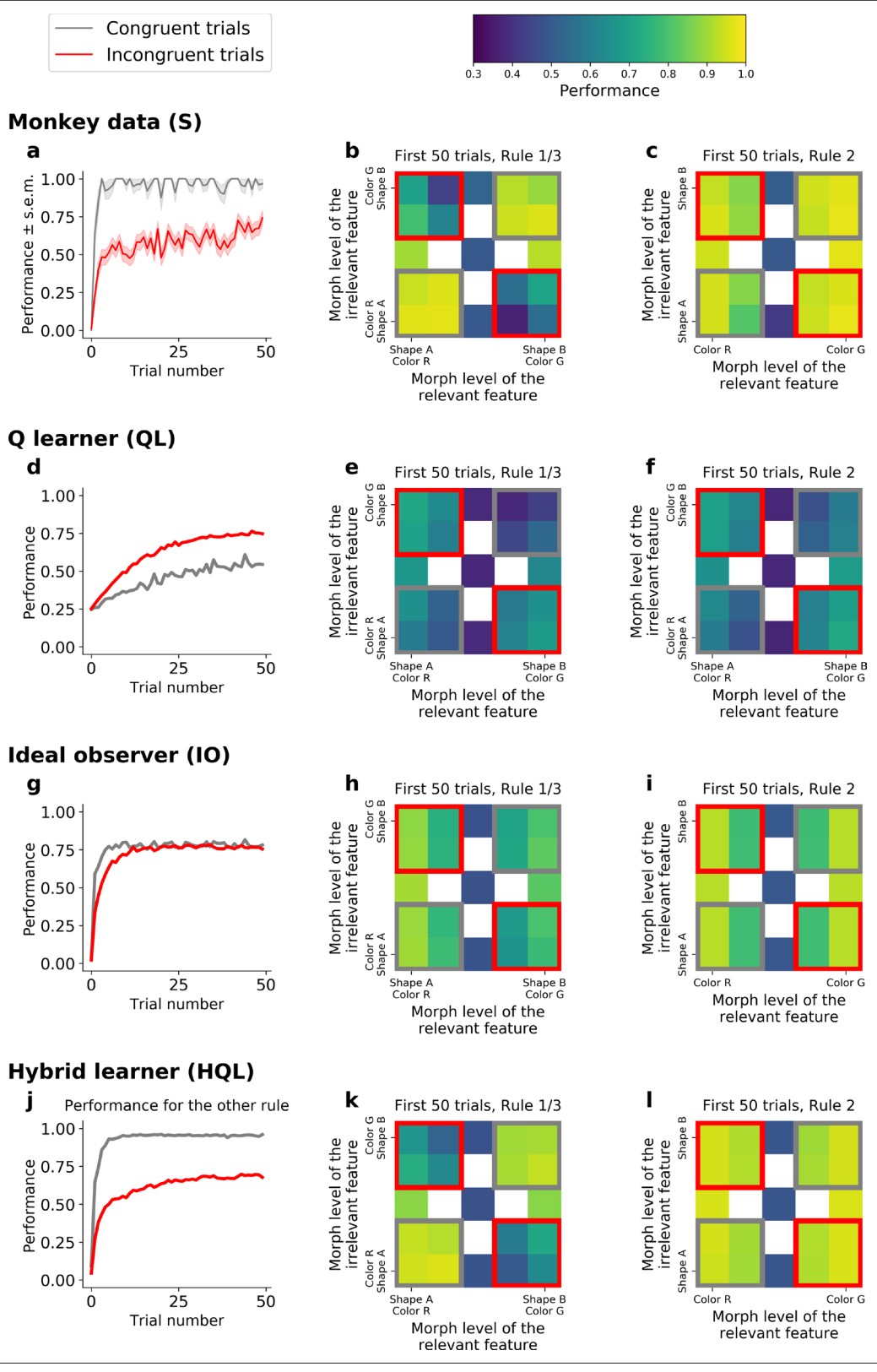

**Figure 5.** Comparison of incongruency effects in Monkey S and behavioral models (QL, IO, and HQL models). (**a**) Performance as a function of trial number for Rules 1 and 3 (combined), for congruent and incongruent trials. (**b**) Performance for Rules 1 and 3 (combined, first 50 trials), as a function of the morph level for both color (relevant) and shape (irrelevant) features. Gray boxes highlight congruent stimuli, red boxes highlight incongruent stimuli.

*Figure 5 continued on next page*

*Figure 5 continued*

(**c**) Performance for Rule 2, as a function of the morph level for both color (relevant) and shape (irrelevant) features. Note the lack of an incongruency effect. (**d–f**) Same as a–c but for the QL model. (**g–i**) Same as a–c for the IO model. (**j–l**) Same as a–c but for the HQL model.

The online version of this article includes the following figure supplement(s) for figure 5:

**Figure supplement 1.** Comparison of incongruency effects in Monkey C and behavioral models (QL, IO, and HQL models).

**Figure supplement 2.** Incongruency effect.

**Figure supplement 3.** Reaction times.

**Figure supplement 4.** Choice probabilities for the three models fitted on Monkey S.

**Figure supplement 5.** Behavior across days.

## Discussion

In the present study, we investigated rule learning in two monkeys trained to switch between three category-response rules. Critically, the animals were not instructed as to which rule was in effect (only that the rule had changed). We compared two classes of models that were able to perform the task: incremental learning and inferential rule switching. Our results suggested that neither model fit the animals' performance well. Incremental learning was too slow to capture the monkeys' rapid learning of the response axis after a block switch. It was also unable to explain the high behavioral performance on Rule 2, which was the only rule requiring responses along the second axis. On the other hand, inferential learning was unable to reproduce the difference in performance for the two rules that required attending to the same feature of the stimulus (color), but responding on different axes (Rules 2 and 3). Finally, when considered separately, neither of these two classes of models could explain the monkeys' difficulty in responding to stimuli that had incongruent responses on the same axes (Rules 1 and 3). Instead, we found that a hybrid model that inferred axes quickly and relearned features slowly was able to capture the monkeys' behavior. This suggests the animals were learning the current axis of response using fast inference while continuously re-estimating the stimulus–response mappings within an axis.

We assessed the generative performance of the hybrid model and falsified (*Palminteri et al., 2017*) incremental learning and inferential rule switching considered separately. The superior explanatory power of the hybrid model suggests that the animals performed both rule switching and rule learning – even in a well-trained regime in which they could, in principle, have discovered perfect rule knowledge. The model suggests that the monkeys discovered only two latent states (corresponding to the two axes of response) instead of the three rules we designed, forcing them to perpetually relearn Rules 1 and 3. These two latent states effectively encode Rule 2 (alone on its response axis) on the one hand, and a combination of Rules 1 and 3 (sharing a response axis) on the other hand. The combination of rules in the second latent state caused the monkeys to continuously update their knowledge of the rules' contingencies (mapping different stimulus features to actions). At first glance, one might be concerned that our major empirical finding about the discrepancy between fast switching and the slow updating of rules was inherent to the task structure. Indeed, a block switch predictably corresponds to a switch of axis, but does not always switch the relevant stimulus feature. Moreover, the features were themselves morphed, creating ambiguity when trying to categorize them and use past rewards for feature discovery. We can thus expect inherently the slower learning of the relevant feature. However, our experiments with IO models demonstrate that an inference process reflecting the different noise properties of axes and features cannot by itself explain the two timescales of learning. The key insight is that, if slow learning about features is simply driven by their inherent noisiness, then the asymptotic performance level with these features should reflect the same degree of ambiguity. However, these do not match. This indicates that the observed fast and slow learning was not a mere representation of the generative model of the task. One caveat about this interpretation is that it assumes that other factors governing asymptotic performance (e.g., motivation or attention, which we do not explicitly control) are comparable between Rule 2 versus Rules 1 and 3 blocks.

The particular noise properties of the task may, however, shed light on a related question raised by our account: why the animals failed to discover the correct three-rule structure, which would clearly

support better performance in Rule 1 and 3 blocks. We believe their failure to do so was not merely a function of insufficient training but would have remained stable even with more practice, as we trained the monkeys until their behavior was stable (and we verified the lack of significant trends in the behavior across days, *Figure 5—figure supplement 5*). Their failure to do so could shed light on the brain's mechanisms for representing task properties and for discovering, splitting, or differentiating different latent states on the basis of their differing stimulus–action–response contingencies. One possible explanation is that the overlap between Rules 1 and 3 (sharing an axis of response) makes them harder to differentiate than Rule 2. In particular, the axis is the most discriminatory feature (being discrete and also under the monkey's own explicit control), whereas the stimulus–reward mappings are noisier and continuous. Alternatively, a two-latent state regime may be rational given the cognitive demands of this task, including the cost of control or constraints on working memory when routinely switching between latent states (stability–flexibility trade-off) (*Musslick and Cohen, 2021*; *Nassar and Troiani, 2021*; *Musslick et al., 2018*).

Thus, all this suggests that the particulars of the behavior may well depend on the details of the task or training protocol. For instance, the monkeys may have encoded each rule as a separate latent state (and shown only fast learning asymptotically) with a different task (e.g., including the missing fourth rule so as to counterbalance axes with stimuli) or training protocol (e.g., longer training, random presentation of the three rules with equal probability of occurrence instead of Rule 2 being interleaved and occurring twice more often, a higher ratio of incongruent versus congruent trials, or less morphed and more prototyped stimuli). Another interesting possibility, which requires future experiments to rule out, is that the disadvantage for Rules 1 and 3 arises not because their sharing response axis necessitates continual relearning, but instead because this sharing per se causes some sort of interference. This alternative view suggests that the relative disadvantage would persist even in a modified design in which the rule was explicitly signaled at each trial, making inference unnecessary.

Understanding how the brain discovers and manipulates latent states would give insight into how the brain avoids catastrophic interference. In artificial neural networks, sequentially learning tasks causes catastrophic interference, such that the new task interferes (overwrites) the representation of the previously learned task (*McCloskey and Cohen, 1989*). In our task, animals partially avoided catastrophic interference by creating two latent states where learning was independent. For example, learning stimulus–response mappings for Rules 1 and 3 did not interfere with the representation of Rule 2. In contrast, Rules 1 and 3 did interfere – behavior was re-learned on each block. Several solutions to this problem have been proposed in machine learning literature such as orthogonal subspaces (*Duncker et al., 2020*) and generative replay (*van de Ven and Tolias, 2019*). Similarly, recent advances in deep reinforcement learning have started to elucidate the importance of incorporating metalearning in order to speed up learning and to avoid catastrophic interference (*Botvinick et al., 2019*; *Hadsell et al., 2020*). Our results suggest the brain might solve this problem by creating separate latent states where learning is possible within each latent state. How these latent states are instantiated in the brain is an open question, and discovering those computations promises exciting new insights for algorithms of learning.

Finally, our characterization of the computational contributions of rule switching and rule learning, and the fortuitous ability to observe both interacting in a single task, leads to a number of testable predictions about their neural interactions. First, our results make the strong prediction that there should be two latent states represented in the brain – the representation for the two rules competing on one axis (Rules 1 and 3) should be more similar to one another than to the neural representation of the rule alone on the other axis (Rule 2). This would not be the case if the neural activity was instead representing three latent causes. Furthermore, our hybrid model suggests there may be a functional dissociation for rule switching and rule learning, such that they are represented in distinct networks. One hypothesis is that this dissociation is between cortical and subcortical regions. Prefrontal cortex may carry information about the animal's trial beliefs (i.e., over the two latent states) in a similar manner as perceptual decision making when accumulating evidence from noisy stimuli (*Gold and Shadlen, 2007*; *Shadlen and Kiani, 2013*; *Rao, 2010*; *Beck et al., 2008*). Basal ganglia may, in turn, be engaged in the learning of rule-specific associations (*Daw and O'Doherty, 2014*; *Daw and Shohamy, 2008*; *Daw and Tobler, 2014*; *Dolan and Dayan, 2013*; *Doya, 2007*; *O'Doherty et al., 2004*; *O'Reilly and Frank, 2006*; *Rescorla, 1988*; *Schultz et al., 1997*; *Yin and Knowlton, 2006*). Alternatively, despite their functional dissociation, future

work may find both rule switching and rule learning are represented in the same brain regions (e.g., prefrontal cortex).

## Materials and methods

### Experimental design and model notation

Two adult (8- to 11-year-old) male rhesus macaques (*Macaca mulatta*) participated in a category-response task. Monkeys S and C weighed 12.7 and 10.7 kg, respectively. All experimental procedures were approved by Princeton University Institutional Animal Care and Use Committee (protocol #3055) and were in accordance with the policies and procedures of the National Institutes of Health.

Stimuli were rendering of three dimensional models that were built using POV-Ray and MATLAB (Mathworks). They were presented on a Dell U2413 LCD monitor positioned at a viewing distance of 58 cm. Each stimulus was generated with a morph-level drawn from a circular continuum between two prototype colors *C* (red and green) and two prototype shapes *S* ('bunny' and 'tee'; *Figure 1b*).

$$X_1^2 + X_2^2 + X_3^2 = P^2$$

where *X* is the parameter value in a feature dimension for example *L*, *a*, *b* values in CIELAB color space. Radius (*P*) was chosen such that there was enough visual discriminability between morph levels. Morph levels in shape dimension were built by circular interpolation of the parameters defining the lobes of the first prototype with the parameters defining the corresponding lobes of the second prototype. Morph levels in the color dimension were built by selecting points along photometrically isoluminant circle in CIELAB color space that connected red and green prototype colors. We used percentage to quantify the deviation of each morph level from prototypes (0% and 100%) on the circular space. Morph levels between 0% and 100% correspond to $-\pi$ to 0, and morph levels 100% and 200% correspond to 0 to $\pi$ on the circular space. Morph levels for color and shape dimension were generated at eight levels: 0%, 30%, 50%, 70%, 100%, 130%, 150%, and 170%. 50% morph levels for one feature (color or shape) were only generated for prototypes for the other feature (shape or color, respectively). The total stimulus set consisted of 48 images. By creating a continuum of morphed stimuli, we could independently manipulate stimulus difficulty along each dimension.

Monkeys were trained to perform three different rules $R = \{R_1, R_2, R_3\}$ (*Figure 1c, d*). All of the rules had the same general structure: the monkeys categorized a visual stimulus according to its shape or color, and then responded with a saccade, $a \in Actions$, to one of four locations *Actions* = {Upper-Left,Upper-Right,Lower-Left,Lower-Right} (*Figure 1a*). Each rule required the monkeys to attend-to and categorize either the color or shape feature of the stimulus, and then respond with a saccade along an axis ($A = \{Axis\ 1, Axis\ 2\}$). *Axis 1* corresponded to Upper-Left, Lower-Right locations and *Axis 2* corresponded to Upper-Right, Lower-Left locations. As such, the correct response depended on the rule in effect and stimulus presented during the trial. Rule 1 ($R_1$) required a response on *Axis 1*, to the Upper-Left or Lower-Right locations when the stimulus was categorized as 'bunny' or 'tee', respectively. Rule 2 ($R_2$) required a response on *Axis 2*, to the Upper-Right or Lower-Left locations when the stimulus was categorized as red or green, respectively. Rule 3 ($R_3$) required a response on *Axis 1*, to the Upper-Left or Lower-Right locations when the stimulus was categorized as green or red, respectively. In this way, the three rules were compositional: Rules 2 and 3 shared the response to the same feature of the stimulus (color) but different axes. Similarly, Rules 1 and 3 shared the response to same axis (*Axis 1*), but to different features.

The monkeys initiated each trials by fixating a dot on the center of the screen. During a fixation period (lasting 500–700 ms), the monkeys were required to maintain their gaze within a circle with radius of 3.25 degrees of visual angle around the fixation dot. After the fixation period, the stimulus and all four response locations were displayed simultaneously. The monkeys made their response by breaking fixation and saccading to one of the four response locations. Each response location was 6 degrees of visual angle from the fixation point, located at 45°, 135°, 225°, and 315° degrees relative to vertical. The stimulus diameter was 2.5 degrees of visual angle. The animal's reaction time was taken as the moment of leaving the fixation window, relative to the onset of the stimulus. Trials with a reaction time lower than 150 ms were aborted, and the monkey received a brief timeout. Following a correct response, monkeys were provided with a small reward, while incorrect responses led to a brief timeout ($r \in \{0, 1\}$). Following all trials, there was an inter-trial interval of 2–2.5 s before the next

trial began. The time distributions were adjusted according to task demands and previous literature (*Buschman et al., 2012*).

Note that both Rules 1 and 3 required a response along the same axis (*Axis 1*). Half of the stimuli were 'congruent', such that they led to the same response for both rules (e.g., a green bunny is associated with an Upper-Left response for both rules). The other half of stimuli were 'incongruent', such that they led to different responses for both rules (e.g., a red bunny is associated with a Upper-Left and Lower-Right response, for Rules 1 and 3, respectively). To ensure the animals were performing the rule, incongruent stimuli were presented on 80% of the trials.

Animals followed a single rule for a 'block' of trials. After the animal's behavioral performance on that rule reached a threshold, the task would switch to a new block with a different rule. The switch between blocks was triggered when the monkeys' performance was greater than or equal to 70% on the last 102 trials of Rules 1 and 3 or the last 51 trials of Rule 2. For Monkey S, each performance, for 'morphed' and 'prototype' stimuli independently, had to be above threshold. Monkey C's performance was weaker, and a block switch occurred when the average performance for all stimuli was above threshold. Also, to avoid that Monkey C perseverated on one rule for an extended period on a subset of days, the threshold was reduced to 65% over the last 75 trials for Rules 1 and 3, after 200 or 300 trials. Switches between blocks of trials were cued by a flashing screen, a few drops of juice, and a long time out (50 s). Importantly, the rule switch cue did not indicate the identity of the rule in effect or the upcoming rule. Therefore, the animal still had to infer the current rule based on its history.

Given the limited number of trials performed each day and to simplify the task structure for the monkeys, the axis of response always changed following a block switch. During *Axis 1* blocks, whether Rule 1 or 3 was in effect was pseudo-randomly selected. These blocks were interleaved by *Axis 2* blocks, which were always Rule 2. Pseudo-random selection of Rules 1 and 3 within *Axis 1* blocks was done to ensure the animal performed at least one block of each rule during each session (accomplished by never allowing for three consecutive blocks of the same rule).

As expected, given their behavioral performance, the average block length varied across monkeys: 50–300 trials for Monkey S, and 50–435 trials for Monkey C. Rule 1 and 3 blocks were on average 199 trials for Monkey S and 222 trials for Monkey C. Rule 2 blocks were shorter because they were performed more frequently and were easier given the task structure. They were on average 56 trials for Monkey S and 52 trials for Monkey C. Overall, the behavioral data include 20 days of behavior from Monkey S, with an average of 14 blocks per day, and 15 days for Monkey C, with an average of 6.5 blocks per day.

## Statistical details on the study design

The sample size is two animals, determined by what is typical in the field. All animals were assigned to the same group, with no blinding necessary.

## Additional details on training

Given the complex structure of the task, monkeys were trained for months until they fully learned the structure of the task and they could consistently perform at least five blocks each day. Monkeys learned the structure of the task in multiple steps. They were first trained to hold fixation and to associate stimuli with reward by making saccades to target locations. To begin with shape categorization (Rule 1), monkeys learned to associate monochrome versions of prototype stimuli with two response locations on *Axis 1*. Stimuli were then gradually colored by using an adaptive staircase procedure. To begin training on color categorization (Rule 2), monkeys learned to associate red and green squares with two response locations on *Axis 2*. Prototype stimuli gradually appeared on the square and finally replaced the square using an adaptive staircase procedure. After this stage, monkeys were trained to generalize across morph levels in 5% morph-level steps using an adaptive staircase method until they could generalize up to 20% morph level away from the prototypes, for color and shape features. Rule 3 was added at this stage with a cue (purple screen background). Once the monkey was able to switch between Rules 1 and 3, the cue was gradually faded and finally removed. After monkeys learned to switch between three rules, the morph levels 30% and 40% were introduced. Monkeys S and C were trained for 36 and 60 months, respectively. Behavioral data reported here are part of data acquisition during electrophysiological recording sessions. From this point, only behavioral sessions in which monkeys performed at least five blocks were included for further analysis. In order to encourage

generalization for shape and color features, during non-recording days, monkeys were trained on the larger number of morph levels (0%, 20%, 30%, 40%, 50%, 60%, 70%, 80%, 100%, 120%, 130%, 140%, 150%, 160%, 170%, and 180%).

## Modeling noisy perception of color and shape

All the models studied below model stimulus perception in the same way (***Figure 2—figure supplement 1***), via a signal-detection-theory like account by which objective stimulus features are modeled as magnitudes corrupted by continuously distributed perceptual noise. In particular, the color and shape of each stimulus presented to the animals are either the prototype features $s_{Tc} \in \{red, green\}$ and $s_{Ts} \in \{bunny, tee\}$, or a morphed version of them. The feature continuous spaces are projected onto the unit circle from $-\pi$ to $\pi$, such that each stimulus feature has a unique radius angle, with prototype angles being 0 and $\pi$. The presented stimulus is denoted $s_M = (s_{Mc}, s_{Ms})$. We hypothesize that the monkeys perceive a noisy version of it, denoted $s_K = (s_{Kc}, s_{Ks})$. We model it by drawing two independent samples, one from each of two Von Mises distributions, centered around each feature ($s_{Mc}$ and $s_{Ms}$, for color and shape), and parameterized by the concentrations $\kappa_C$ and $\kappa_S$, respectively. The models estimate each initial feature presented by computing its posterior distribution, given the perceived stimulus, that is by Von Mises distributions centered on $s_{Kc}$ and $s_{Ks}$, with same concentrations $\kappa_C$ and $\kappa_S$.

$$\forall i \in \{c, s\}$$

$$Pr(s_{K_i} \mid s_{M_i}) = VonMises(s_{M_i}, \kappa_i)$$

and so

$$Pr(s_{M_i} \mid s_{K_i}) = VonMises(s_{K_i}, \kappa_i)$$

with

$$VonMises(\mu, \kappa) \propto exp(\kappa(cos(x - \mu)))$$

## Modeling action selection

All the models studied below use the same action-selection stage. Given the perceived stimulus at each trial $s_K = (s_{Kc}, s_{Ks})$, an action is chosen so as to maximize the expected reward $\mathbb{E}(r \mid \mathbf{s_K})$ by computing $\max_a Pr(r = 1 \mid \mathbf{s_K}, a)$ which corresponds to maximizing the probability of getting a reward, given the perceived stimulus. We use the notation $Q(\mathbf{s_K}, a) = \mathbb{E}(r(a) \mid \mathbf{s_K})$ as in previous work (***Dayan and Daw, 2008***; ***Daw et al., 2006***) and refer to these values as $Q$ values. Note that even a deterministic 'max' choice rule at this stage does not, in practice, imply noiseless choices, since $Q$ depends on $s_K$, and the perceptual noise in this quantity gives rise to variability in the maximizing action that is graded in action value, analogous to a softmax rule. For that reason, we do not include a separate choice-stage softmax noise parameter (which would be unidentifiable relative to perceptual noise $\kappa$), though we do nevertheless approximate the max choice rule with a softmax (but using a fixed temperature parameter), for implementational purposes (specifically, to make the choice model differentiable). To control the asymptotic error rate, we also include an additional probability of lapse (equivalently, adding 'epsilon-greedy' choice). Altogether, two fixed parameters implement an epsilon-greedy softmax action-selection rule: the lapse rate $\varepsilon$ and the inverse temperature $\beta$.
The action-selection rule is:

$$\forall a \in Actions$$

$$Pr(a \mid \mathbf{s_K}) = \frac{\epsilon}{4} + (1 - \epsilon) \cdot softmax[Q(\mathbf{s_K}, a)]$$

The lapse rate $\varepsilon$ is directly estimated from the data, by computing the proportion of trials where the incorrect axis of response is chosen, asymptotically. It is evaluated to 0.02 for both Monkeys S and C. The inverse temperature of the softmax $\beta$ is also fixed ($\beta = 10$), to allow the algorithm to approximate the max while remaining differentiable (cf. use of Stan below).

## Fit with Stan

All our models shared common noisy perceptual input and action selection stages (*Figure 2—figure supplement 1*, and Methods). The models however differed in the intervening mechanism for dynamically mapping stimulus to action value (see *Figure 2—figure supplement 1*). Because of noise perception at each trial, and because the cumulative distribution function of a Von Mises is not analytic, the models are fitted with Monte Carlo Markov chains (MCMC) using Stan (*Carpenter, 2017*). Each day of recording is fitted separately, and the mean and standard deviation reported in *Table 1* are between days. We validated convergence by monitoring the potential scale reduction factor R-hat (which was <1.05 for all simulations) and an estimate of the effective sample size (all effective sample sizes >100) of the models' fits (*Gelman and Rubin, 1992*).

Models' plots correspond to an average of 1000 simulations of each day of the dataset (with the same order of stimuli presentation). Statistics reported in the article were done with Fisher's exact test (except a *t*-test for reaction times, *Figure 5—figure supplement 3*).

## Incremental learner: QL model

In this model, the agent is relearning each rule after a block as a mapping between stimuli and actions, by computing a stimulus-action value function as a linear combination of binary feature–response functions $\phi\left(s_K, a\right)$ with feature–response weights **w**. This implements incremental learning while allowing for some generalization across actions. The weights are updated by the delta rule (*Q* learning with linear function approximation; *Sutton and Barto, 2018*). The weights are reset from one block to the other, and the initial values for each reset are set to zero. Fitting them does not change the results (see Parameter values').

## Computation of the feature–response matrix

Given a morph perception $s_K = \left(s_{Kc}, s_{Ks}\right)$ at trial $t$, a feature–response matrix is defined as:

$$\phi(\mathbf{s_K}) = \begin{pmatrix} x_C & 0 & 0 & 0 \\ x_S & 0 & 0 & 0 \\ 0 & x_C & 0 & 0 \\ 0 & x_S & 0 & 0 \\ 0 & 0 & x_C & 0 \\ 0 & 0 & x_S & 0 \\ 0 & 0 & 0 & x_C \\ 0 & 0 & 0 & x_S \end{pmatrix}$$

where $x_C \in \{-1, 1\}$ depends on whether the perceived morph for color $s_{Kc}$ is classified as green or red, and $x_S \in \{-1, 1\}$ whether the perceived morph for shape $s_{Ks}$ is classified as tee or bunny (see 'Modeling noisy perception of color and shape'). For the algorithm to remain differentiable, we approximate {−1,1} with a sum of sigmoids. Each column of the matrix $\phi\left(s_K\right)$ is written $\phi\left(s_K, a\right)$ below and corresponds to an action $a \in Actions$.

## Linear computation of Q values and action selection

In order to compute Q values, the feature–response functions $\phi\left(s_K, a\right)$ are weighted by the feature–response weight vector $w = \left(w_1, .., w_8\right)$ :

$$\forall a \in Actions$$

$$Q\left(s_K, a\right) = w \cdot \phi\left(s_K, a\right)$$

Action selection is done through the epsilon-greedy softmax rule above.
Thus asymptotic learning of Rule 1 would require $w = [0, 1, 0, -1, 0, 0, 0, 0]$. Learning Rule 2 would require $w = [0, 0, 0, 0, -1, 0, 1, 0]$. Learning Rule 3 would require $w = [-1, 0, 1, 0, 0, 0, 0, 0]$.

## Weight vector update

Once an action $a_t$ is chosen and a reward $r_t$ is received at trial $t$, the weights are updated by the delta rule (**Sutton and Barto, 2018**) with learning rate $\alpha$.

$$\mathbf{w} \leftarrow \mathbf{w} + \alpha(r_t - Q(\mathbf{s_K}, a_t))\phi(\mathbf{s_K}, a_t)$$

## Parameter values

$\beta$ and $\varepsilon$ are fixed to, respectively, 10 and 0.02.

Parameters values are reported in **Table 1**. As predicted from the behavior, noise perception is higher for shape than for color ($\kappa_C > \kappa_S$). The initial weight vector $w_0$ is set to zero at the beginning of each block day. Fitting these weights instead gives the same results (as then $w_0$ has a mean of $[-0.070, 0, 0.031, 0.093, -0.054, -0.081, -0.022, 0.062, -0.0048]$ for Monkey S and $w_0$ has a mean of $[-0.099, 0.0070, 0.069, -0.089, -0.053, -0.011, 0.030, -0.0098]$ for Monkey C).

## Optimal Bayesian inference over rules: IO model

In this model, we assume a perfect knowledge of combination mappings between prototype stimuli and actions as *rules*. Learning is discovering which rule is in effect by Bayesian inference. This is done through learning, over the trials, the probability for each rule to be in effect in a block (or *belief*) from the history of stimuli, actions, and rewards. At each trial, this belief is linearly combined to the likelihood of a positive reward given the stimulus to compute a value for each action. This likelihood encapsulates knowledge of the three experimental rules. An action is chosen as per described above (see 'Modeling action selection'). The beliefs over rules are then updated through Bayes rule using the likelihood of the reward received, given the chosen action and the stimulus perception. Once the rule is discovered, potential errors thus only depend on the possible miscategorization of the stimulus features (see 'Modeling noisy perception of color and shape'), or eventually on exploration (see 'Modeling action selection').

## Belief over rules

The posterior probability of rule $R \in \mathfrak{R}$ to be in effect in the block is called the belief over the rule $b(R) = \Pr(R|s_K, a, r)$ given the perceived stimulus $s_K$, the action $a$ and the reward $r$. The beliefs $b(R)$ at the beginning of each block are initialized to $b_0 = [b_1, 1 - b_1 - b_3, b_3]$ where $b_1$ and $b_3$ are fitted, to test for a systematic initial bias toward one rule.

## Computation of values

The beliefs are used to compute the Q values for the trial:

$$\forall a \in Actions$$
$$Pr(r = 1 \mid \mathbf{s_K}, a) = \sum_{R \in \mathcal{R}} Pr(r = 1, R \mid \mathbf{s_K}, a) = \sum_{R \in \mathcal{R}} Pr(r = 1 \mid \mathbf{s_K}, a, R) \cdot b(R)$$

with (marginalization over the possible morph stimuli presented):

$$\forall R \in \mathcal{R}$$
$$Pr(r \mid \mathbf{s_K}, a, R) = \sum_{\mathbf{s_M}} Pr(r, \mathbf{s_M} \mid \mathbf{s_K}, a, R) = \sum_{\mathbf{s_M}} Pr(r \mid \mathbf{s_M}, a, R) Pr(\mathbf{s_M} \mid \mathbf{s_K})$$

Noting $p_C = p(s_{Mc} = red \mid s_{Kc})$ and $p_S = p(s_{Ms} = bunny \mid s_{Ks})$ gives:

$$Q(\mathbf{s_K}, a = Upper - Left) = p_S \cdot b(R_1) + (1 - p_C) \cdot b(R_3)$$
$$Q(\mathbf{s_K}, a = Upper - Right) = (1 - p_S) \cdot b(R_1) + p_C \cdot b(R_3)$$
$$Q(\mathbf{s_K}, a = Lower - Left) = (1 - p_C) \cdot b(R_2)$$
$$Q(\mathbf{s_K}, a = Lower - Right) = p_C \cdot b(R_2)$$

## Belief update

From making an action $a_t \in Actions$, the agent receives a reward $r_t \in \{0, 1\}$, and the beliefs over rules are updated:

$$\forall R \in \mathcal{R}$$

$$b(R) \leftarrow Pr(r_t \mid \mathbf{s_K}, a_t, R) \cdot b(R)$$

with $Pr(r_t \mid \mathbf{s_K}, a_t, R)$ the likelihood of observing reward $r_t$ for the chosen action $a_t$.

Note that because of the symmetry of the task, $Pr(\neg r_t \mid \mathbf{s_K}, a_t, R) = 1 - Pr(r_t \mid \mathbf{s_K}, a_t, R)$ .

### Parameter values

$\beta$ and $\varepsilon$ are, respectively, fixed to 10 and 0.02.

Parameter values are reported in *Table 1*. As predicted from the behavior, there is an initial bias for Rule 2 for the model fitted on Monkey S behavior ($b_2 > b_3 > b_1$). Also, noise perception is higher for shape than for color for both monkeys ($\kappa_C > \kappa_S$). In the version of the model with low perceptual color noise (*Figure 3* and *Figure 3—figure supplement 1*), all the parameters remain the same, except that we fix $\kappa_C = 6$ for all simulated days.

### Hybrid incremental learner: HQL model

The hybrid incremental learner combines inference over axes with incremental learning, using a $Q$ learning with function approximation to relearn the likelihood of rewards given stimuli per axis of response.

### Belief over axes

The posterior probability of an axis $A \in \mathcal{A}$ to be the correct axis of response in a block is called the belief over axis $b(A) = Pr(A \mid \mathbf{s_K}, a, r)$, given the perceived stimulus $s_K$ , the action $a$ and the reward $r$.

The beliefs over axes are initialized at the beginning of each block to $b_0 = (b_{ax}, 1 - b_{ax})$ .

### Computation of the feature–response matrix

As for the incremental learner above, given a morph perception $s_K = (s_{Kc}, s_{Ks})$ at trial $t$, a feature–response matrix is defined as:

$$\phi(\mathbf{s_K}) = \begin{pmatrix} x_C & 0 & 0 & 0 \\ x_S & 0 & 0 & 0 \\ 0 & x_C & 0 & 0 \\ 0 & x_S & 0 & 0 \\ 0 & 0 & x_C & 0 \\ 0 & 0 & x_S & 0 \\ 0 & 0 & 0 & x_C \\ 0 & 0 & 0 & x_S \end{pmatrix}$$

where $x_C \in \{-1, 1\}$ depends on whether the perceived morph for color $s_{Kc}$ is classified as green or red, and $x_S \in \{-1, 1\}$ whether the perceived morph for shape $s_{Ks}$ is classified as tee or bunny (see 'Modeling noisy perception of color and shape'). Each column of the matrix $\phi(s_K)$ is written $\phi(s_K, a)$ below and corresponds to an action $a \in Actions$.

### Computation of values

The beliefs are used to compute the $Q$ values for the trial:

$$\forall a \in Actions$$

$$Q(\mathbf{s_K}, a) = Pr(r = 1 \mid \mathbf{s_K}, a) = \sum_{A \in \mathcal{A}} Pr(r = 1, A \mid \mathbf{s_K}, a) = \sum_{A \in \mathcal{A}} Pr(r = 1 \mid \mathbf{s_K}, a, A) \cdot b(A)$$

Contrary to the ideal observer, here the likelihood of reward per action $Pr(r = 1 \mid \mathbf{s_K}, a, A)$ is learned through function approximation.

$$Pr(r = 1 \mid \mathbf{s_K}, a, A) = sigmoid(\mathbf{w} \cdot \phi(\mathbf{s_K}, a))$$

Action selection is done through the epsilon-greedy softmax rule.

## Weight vector update

Once an action $a_t$ is chosen and a reward $r_t$ is received at trial $t$, the weights are updated through gradient descent with learning rate $\alpha$.

$$p_t = Pr(r = 1 \mid \mathbf{s_K}, a_t, A_t)$$

$$w \longleftarrow w + \alpha \left(r_t - p_t\right) \phi \left(s_K, a\right) p_t \left(1 - p_t\right)$$

As learning improved steadily in this model contrary to the asymptotic behavior of monkeys, we implemented a weight decay to asymptotic values $w_0$:

$$w \longleftarrow \left(1 - \eta\right) w + \eta w_0$$

Note that resetting the weights at the beginning of each block and adding a weight decay (or a learning rate decay) provide similar fits to the dataset. Also, this decay can be included in the previous two models without any change of our results and conclusions.

## Belief update

From making an action $a_t \in Actions$, the agent receives a reward $r_t \in \left\{0, 1\right\}$ and the beliefs over axes are updated:

$$\forall A \in \mathcal{A}$$

$$b(A) \leftarrow Pr(r_t \mid \mathbf{s_K}, a_t, A) \cdot b(A)$$

## Parameter values

$\beta$ and $\varepsilon$ are fixed to, respectively, 10 and 0.02. As predicted from the behavior, noise perception is higher for shape than for color for both monkeys ($\kappa_C > \kappa_S$). Also, the model fitted on Monkey S behavior has an initial bias for *Axis 2* ($b_{ax} < 0.5$). For fitting the model on Monkey C behavior, we fix $b_{ax} = 0.5$. Finally, the fitted values of $w_0$ correspond to an encoding of an average between Rules 1 and 3 on *Axis 1*, and an encoding of Rule 2 on *Axis 2*, for both monkeys (see *Table 1*).

## Research standards

Codes and data supporting the findings of this study are available on GitHub (https://github.com/buschman-lab/FastRuleSwitchingSlowRuleUpdating, copy archived at swh:1:rev:9a7cde-4a06e8571d7b955750b599221c40acfac5; *Bouchacourt, 2022*).

| Resource | Source | Identifier |
|---|---|---|
| *Macaca mulatta* | Mannheimer Foundation | 10-52,10-153 |
| PyStan 2.19 | Stan Development Team | https://mc-stan.org/users/interfaces/pystan.html |
| POV-ray | Persistence of Vision Pty Ltd | http://www.povray.org/ |
| MATLAB R2015a | Mathworks | https://www.mathworks.com |
| Python 3.6 | Python software foundation | https://www.python.org/ |

# Acknowledgements

The authors thank Sam Zorowitz for helpful discussions on the statistical modeling platform Stan.

# Additional information

## Funding

| Funder | Grant reference number | Author |
|---|---|---|
| U.S. Army Research Office | ARO W911NF-16-1-047 | Nathaniel D Daw |

| Funder | Grant reference number | Author |
|---|---|---|
| NIMH | R01MH129492 | Timothy J Buschman |

The funders had no role in study design, data collection, and interpretation, or the decision to submit the work for publication.

## Author contributions

Flora Bouchacourt, Conceptualization, Software, Formal analysis, Validation, Investigation, Visualization, Methodology, Writing - original draft, Writing – review and editing; Sina Tafazoli, Conceptualization, Data curation, Validation, Investigation, Methodology, Writing – review and editing; Marcelo G Mattar, Conceptualization, Validation, Methodology, Writing – review and editing; Timothy J Buschman, Conceptualization, Resources, Data curation, Supervision, Funding acquisition, Validation, Methodology, Project administration, Writing – review and editing; Nathaniel D Daw, Conceptualization, Supervision, Funding acquisition, Validation, Methodology, Project administration, Writing – review and editing

## Author ORCIDs

Flora Bouchacourt ⓘ http://orcid.org/0000-0002-8893-0143
Sina Tafazoli ⓘ http://orcid.org/0000-0003-1926-0227
Marcelo G Mattar ⓘ http://orcid.org/0000-0003-3303-2490
Timothy J Buschman ⓘ http://orcid.org/0000-0003-1298-2761
Nathaniel D Daw ⓘ http://orcid.org/0000-0001-5029-1430

## Ethics

All experimental procedures were approved by Princeton University Institutional Animal Care and Use Committee (protocol #3055) and were in accordance with the policies and procedures of the National Institutes of Health.

## Decision letter and Author response

Decision letter https://doi.org/10.7554/eLife.82531.sa1
Author response https://doi.org/10.7554/eLife.82531.sa2

# Additional files

## Supplementary files

• MDAR checklist

## Data availability

Codes and data supporting the findings of this study are available on GitHub (https://github.com/buschman-lab/FastRuleSwitchingSlowRuleUpdating, copy archived at swh:1:rev:9a7cde4a06e8571d7b955750b599221c40acfac5).

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
