## [Editor Report]

This important study modeled monkeys' behavior in a stimulus-response rule-learning task to show that animals can adopt mixed strategies involving inference for learning latent states and incremental updating for learning action-outcome associations. The task is cleverly designed, the modeling is rigorous, and importantly there are clear distinctions in the behavior generated by different models, which makes the conclusions convincing.

---

## [Decision Letter]

**Decision letter after peer review:**

Thank you for submitting your article "Fast rule switching and slow rule updating in a perceptual categorization task" for consideration by *eLife*. Your article has been reviewed by 2 peer reviewers, and the evaluation has been overseen by a David Badre as Reviewing Editor and Michael Frank as the Senior Editor. The reviewers have opted to remain anonymous.

Essential revisions:

The reviewers and editors were in agreement that this is a strong contribution. In consultation, it was agreed that to strengthen the impact, there are some points that could be revised. In particular:

1) The design should be clarified to address some points of confusion about the specifics.

2) Both reviewers raised some alternatives that should be considered. And, if not ruled out, they should be discussed in relation to the findings and in comparison to the favored account.

These points are detailed in the reviewers' comments below. We have left them unedited as they are clear in the points that could be strengthened through revision.

*Reviewer #1 (Recommendations for the authors):*

As indicated in the discussion, the regularity in the stimulus-response mappings makes it possible for the animals to learn the task structure and behave according to the ideal observer model, which would be an optimal strategy. However, they didn't fully accomplish this, likely due to the difficulty of learning the full task structure compared to the repeated switches between response sets. On one hand, settling on a hybrid strategy might be the subjects' preferred trade-off between effort and optimal outcomes. The authors seem to favor this as a possible explanation . However, it could also be that monkeys would eventually learn to use rule inference with more practice. With this in mind, it would be helpful if more detail about the duration of the monkeys' training in the task could be supplied in the methods. It would also be of interest to know whether there are any changes in behavior across sessions included in this study.

*Reviewer #2 (Recommendations for the authors):*

1. I found the introduction to be a little confusing. Reference is made obliquely to a "general purpose incremental learning mechanism" but I was initially a bit unclear about what is meant here. It would have been helpful for me as a reader if the authors had used more standard terminology, e.g. introducing trial and error learning and latent state inference; and highlighting explicitly work that has focussed on understanding how these processes are implemented, and how they work together, and especially highlighting work in monkeys. There is quite a lot of reference to RNN models and/or meta-learning methods but these seem less relevant here given the modelling approach. Generally, I thought the introduction could have been clearer, more focused, and spoken more directly to the specific questions addressed in the paper.

2. Although the paper is generally quite clear, I found the description of the task a little bit hard to follow initially. In particular, the essential feature of the design – that rule 2 occurs twice as often as the other rules and rules 1 and 3 never occur in sequence – is visible in Figure 1c but is otherwise not explicitly mentioned in the main text (or was not sufficiently highlighted for me). This is not an incidental feature of the task ("to facilitate learning and performance") but is the key driver of the main result, so I think it needs a bit more introduction/foregrounding for reader clarity.

---

## [Author Response]

Essential revisions:The reviewers and editors were in agreement that this is a strong contribution. In consultation, it was agreed that to strengthen the impact, there are some points that could be revised. In particular:1) The design should be clarified to address some points of confusion about the specifics.2) Both reviewers raised some alternatives that should be considered. And, if not ruled out, they should be discussed in relation to the findings and in comparison to the favored account.These points are detailed in the reviewers' comments below. We have left them unedited as they are clear in the points that could be strengthened through revision.

We are grateful to the editors and the reviewers for the thoughtful comments. As discussed below, we have addressed the suggested revisions in the new version.

Reviewer #1 (Recommendations for the authors):As indicated in the discussion, the regularity in the stimulus-response mappings makes it possible for the animals to learn the task structure and behave according to the ideal observer model, which would be an optimal strategy. However, they didn't fully accomplish this, likely due to the difficulty of learning the full task structure compared to the repeated switches between response sets. On one hand, settling on a hybrid strategy might be the subjects' preferred trade-off between effort and optimal outcomes. The authors seem to favor this as a possible explanation. However, it could also be that monkeys would eventually learn to use rule inference with more practice. With this in mind, it would be helpful if more detail about the duration of the monkeys' training in the task could be supplied in the methods. It would also be of interest to know whether there are any changes in behavior across sessions included in this study.

We agree (and now make clear in the discussion) that the monkeys’ strategies might be different depending on the details of training. Regarding reporting what was actually done here, the Methods paragraph “Additional details on training” reports the length and protocol used for training the two monkeys. As mentioned in the methods, our intent was to train the monkeys until their behavior was stable. Consistent with this, in response to the reviewer’s comment, we examined this formally but found no significant trends in the averaged performance across days (Figure 5 —figure supplement 5a,d), or in the per day-by-day estimated model parameters (noise perception and learning rate Figure 5 —figure supplement 5b,c,e), with the exception of the learning rate in Monkey C which was moderately significant (Figure 5 —figure supplement 5f, though not surviving multiple comparison correction). We now discuss this in the manuscript.

Reviewer #2 (Recommendations for the authors):1. I found the introduction to be a little confusing. Reference is made obliquely to a "general purpose incremental learning mechanism" but I was initially a bit unclear about what is meant here. It would have been helpful for me as a reader if the authors had used more standard terminology, e.g. introducing trial and error learning and latent state inference; and highlighting explicitly work that has focussed on understanding how these processes are implemented, and how they work together, and especially highlighting work in monkeys. There is quite a lot of reference to RNN models and/or meta-learning methods but these seem less relevant here given the modelling approach. Generally, I thought the introduction could have been clearer, more focused, and spoken more directly to the specific questions addressed in the paper.

We have streamlined and revised the introduction with the reviewer’s points and terminology in mind.

2. Although the paper is generally quite clear, I found the description of the task a little bit hard to follow initially. In particular, the essential feature of the design – that rule 2 occurs twice as often as the other rules and rules 1 and 3 never occur in sequence – is visible in Figure 1c but is otherwise not explicitly mentioned in the main text (or was not sufficiently highlighted for me). This is not an incidental feature of the task ("to facilitate learning and performance") but is the key driver of the main result, so I think it needs a bit more introduction/foregrounding for reader clarity.

We thank the reviewer for this comment and have highlighted this essential feature of the design in the paragraph “Task design and performance”.